# Evo-NeRF: Evolving NeRF for Sequential Robot Grasping of Transparent Objects

**Justin Kerr, Letian Fu, Huang Huang, Yahav Avigal, Matthew Tancik, Jeffrey Ichnowski, Angjoo Kanazawa, Ken Goldberg**

University of California, Berkeley

**Abstract:** Sequential robot grasping of transparent objects, where a robot removes objects one by one from a workspace, is important in many industrial and household scenarios. We propose Evolving NeRF (Evo-NeRF), leveraging recent speedups in NeRF training and further extending it to rapidly train the NeRF representation concurrently to image capturing. Evo-NeRF terminates training early when a sufficient task confidence is achieved, evolves the NeRF weights from grasp to grasp to rapidly adapt to object removal, and applies additional geometry regularizations to make the reconstruction smoother and faster. General purpose grasp planners such as Dex-Net may underperform with NeRF outputs because there can be unreliable geometry from rapidly trained NeRFs. To mitigate this distribution shift, we propose a Radiance-Adjusted Grasp Network (RAG-Net), a grasping network adapted to NeRF's characteristics through training on depth rendered from NeRFs of synthetic scenes. In experiments, a physical YuMi robot using Evo-NeRF and RAG-Net achieves a 89% grasp success rate over 27 trials on single objects, with early capture termination providing a 41% speed improvement with no loss in reliability. In a sequential grasping task on 6 scenes, Evo-NeRF reusing network weights clears 72% of the objects, retaining similar performance as reconstructing the NeRF from scratch (76%) but using 61% less sensing time. See https://sites.google.com/view/evo-nerf for more materials.

## 1 Introduction

Sequentially grasping transparent objects is critical in industry, pharmaceuticals and households. Sensing these objects is difficult; since camera-based sensors see through transparent objects from most angles, assumptions underlying traditional disparity and structure-from-motion-based methods break. Data driven approaches rely on large synthetic and real-world datasets to address this problem. ClearGrasp [1] trains a CNN to infer local surface normals on transparent objects from RGBD images based on Blender synthetic examples. They show impressive results on 3-5 transparent objects separated by 2cm, but note challenges with open-top containers, partial occlusions in clutter, background distractors, and transparent objects' shadows.

Neural Radiance Fields (NeRFs) [2] are a 3D representation originally designed for novel view synthesis which can reconstruct traditionally challenging-to-model scenes that include transparent objects. Dex-NeRF [3] uses NeRF to grasp transparent objects, but costs hours of computation per grasp. Recent dramatic advancements in NeRF training speed have opened the door for real-time usage [4, 5, 6]. We propose Evo-NeRF, a method for rapidly training NeRF for grasping, and RAG-Net, a neural network for robustly computing grasps from NeRF rendered depth images. We apply Evo-NeRF in a purely online setting to sequentially grasp transparent objects in clutter in the time-span of 10s of seconds, as required in dish loading, table clearing, and other household tasks.

To make NeRF practical for robotic grasping, we build on Instant-NGP [6], a fast variant of NeRF. Rather than training on a fixed set of images, we incrementally optimize over a stream of images as they are captured during a robot motion. Due to NeRF's varying convergence speed on different difficulty scenes, we propose a method to terminate image capture upon achieving sufficient task

---

Correspondence to justin_kerr@berkeley.edu

6th Conference on Robot Learning (CoRL 2022), Auckland, New Zealand.

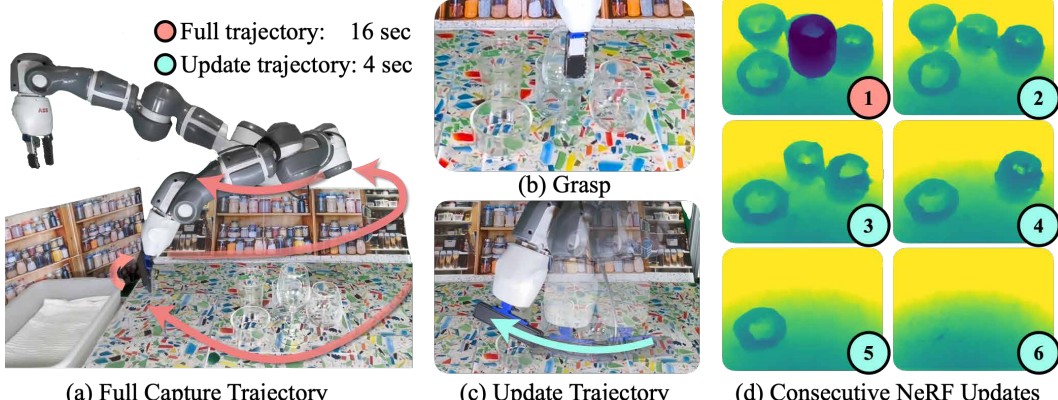

Figure 1: **Sequential object removal.** **(a)** The YuMi moves the camera through a hemisphere trajectory (red arrow) to capture a scene of 5 glass objects, training a NeRF simultaneously. **(b)** The robot immediately plans and executes a grasp from the NeRF after camera capture **(c)** Short camera trajectories are used to evolve the NeRF between grasps **(d)** Evo-NeRF first reconstructs the whole scene (1) with the camera trajectory shown in (a), then progressively updates the scene with small camera captures shown in (c) as objects are removed.

confidence. We further adapt NeRF to sequential grasping by re-using NeRF weights from grasp to grasp and demonstrate its rapid adaptability to object removal. Since we propose that the robot captures images and trains a NeRF as it moves, motion blur, kinematic limitations, and speed considerations reduce the quality of the recovered geometry and introduce prominent spurious geometry known as *floaters*. We propose adding geometry regularization to the training objective, which improves the recovered geometry, but out-of-the-box grasp planners still struggle to find quality grasps due to remaining artifacts. Dex-NeRF [3] on the other hand, could use an out-of-the-box grasp sampler because it used diverse, high-quality, calibrated, *still* images captured in an offline process.

To mitigate the lower-quality NeRF reconstructions, we propose a novel pipeline to train a grasping network on depth maps directly from NeRFs, which are trained on photorealistic renderings of transparent objects. We find the grasping network transfers well to real-world NeRF reconstructions. This pipeline utilize the training speed of Instant-NGP—without it, the pipeline would be computationally infeasible. Real robot experiments using an actuatable camera to capture images suggest that Evo-NeRF can reconstruct graspable scene geometry rapidly and reliably, when combined with RAG-Net achieving a 89% success rate on single objects within *9.5* seconds of image capturing.

The contributions of this paper are: (1) novel usage of NeRF in a sequential setting, rapidly evolving the NeRF representation between grasps, (2) improvements in scene geometry reconstruction speed built on existing methods, (3) an approach based on task confidence to efficiently stop image capturing early, (4) a novel training pipeline in simulation to acclimate a grasping planner to NeRF geometry characteristics, (5) a dataset of 8667 Blender rendered scenes of transparent objects with robust grasps, and (6) experimental data suggesting that Evo-NeRF enables rapid grasping on NeRF.

## 2  Related Work

**Neural Radiance Fields (NeRF)**  NeRF [2] is a neural-network scene representation that enables photorealistic synthesis of novel views of a scene given a set of images and camera matrices. The representation is a function of location and view angle, and returns a density and view-dependent color. Densities and colors sampled along a camera ray are aggregated using volumetric rendering to produce a pixel color. NeRF is popular in the computer vision and graphics communities with the applications in dynamic scene reconstruction [7, 8], image synthesis [9, 10, 11], pose estimation [12, 13, 14], and more. Optimizing NeRF to reconstruct a single scene can take hours or days—making it impractical for many robotics applications. Instant-NGP [6] and others [4, 15] speed up NeRF by using voxel feature grids instead of multi-layer perceptions to simplify or remove [5] a computational bottleneck. We build on Instant-NGP [6], which uses a learnable hash encoding and highly optimized CUDA implementation to speed up NeRF training from the order of hours to seconds. Others have also sped up NeRF by reusing computation between scenes by utilizing priors. Existing methods [16, 17, 18, 19, 20] use convolutional neural networks (CNNs) to extract image features as input to a shared network that predicts the NeRF. Tancik et al. [21] and

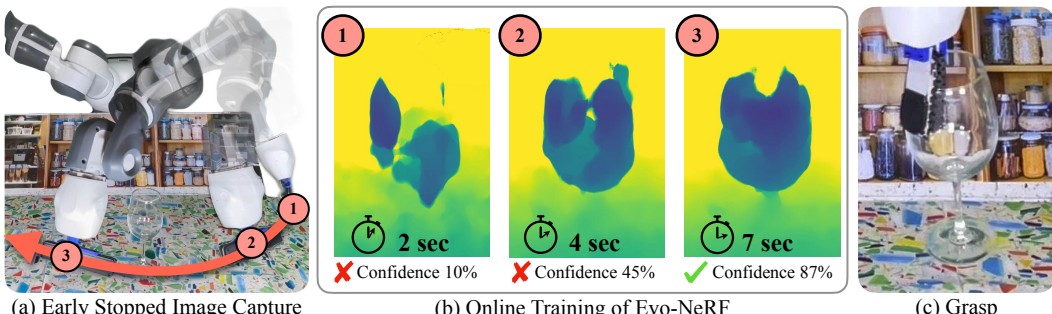

| (a) Early Stopped Image Capture | (b) Online Training of Evo-NeRF | (c) Grasp |

Figure 2: **Evo-NeRF for rapid grasping: (a)** The robot begins capturing images along a hemisphere trajectory (red arrow) **(b)** Evo-NeRF trains a NeRF during arm motion, building graspable geometry of the wineglass. Grasp confidence from RAG-Net builds as NeRF learns geometry, reaching the stopping threshold at (3). **(c)** The robot executes the grasp predicted by RAG-Net at the early stop point.

Gao et al. [22] speed up NeRF training using meta-learning to initialize network weights to ones that converge faster for likely scenes. In this work, we propose using past reconstructions of a scene as an initialization for the current reconstruction, allowing rapid adaptation to changes in the scene.

**NeRFs in Robotics**  Recent research has shown NeRFs to be a promising scene representation for downstream robotics tasks such as navigation and SLAM [23, 24, 25] and manipulation [26, 3, 27]. Yen-Chen et al. [12] and Tseng et al. [28] use a trained NeRF model to estimate an object's 6-DOF pose by minimizing the residuals between a rendered image and a given observed image. Driess et al. [29] train a graph neural network to learn a dynamics model in a multi-object scene represented through a NeRF model, while Li et al. [26] condition a NeRF model on a learned latent dynamics model to plan to visual goals in simulated environments. We propose building on advances in NeRF and its applications to robotics to speed up NeRF-based grasping for practical uses.

**Grasping Transparent Objects:** Most closely related to this paper are two recent works leveraging NeRFs to manipulate objects that cannot be detected by commodity RGBD sensors. Yen-Chen et al. [27] use a NeRF model offline to train dense object descriptors and manipulate thin and reflective objects. Ichnowski et al. [3] show that manually constructing an offline dataset of a given scene then training NeRF allows off-the-shelf grasp planners [30] to compute successful grasps on transparent objects. ClearGrasp [1] trains a Sim2Real depth prediction network on RGB images, then uses this network in real environments to estimate surface geometry for grasps. This idea has been extended to pointclouds and with more efficient real-world data collection [31, 32]. GraspNeRF [33] explores neural rendering as supervision to train a multi-view feature volume network similar to Kar et al. [34] on photorealistic simulated scenes, which is used for grasping. In contrast, using NeRF directly in real-time does not require a prior on the scene at hand for reconstruction, and has superior performance on thin surfaces, occlusions, and complex backgrounds.

## 3   Problem Statement

Given a set of transparent objects resting on a planar workspace, the objective is for the robot to find, grasp, and remove each object quickly. Objects are placed close to each other (2.5 cm) and the robot has an actuatable camera and a parallel jaw gripper (Fig. 1). The focus is on finding robust grasps rapidly, with grasp success measured as transporting an object without dropping.

We assume (1) objects rest in graspable stable poses on a flat surface, (2) objects are in the reachable workspace of the robot with a known forward kinematic model, (3) the camera-to-arm transform is known and stable, and (4) the robot can follow a known obstacle-free trajectory to capture images.

## 4   Method

To rapidly compute robust grasps, we propose *Evo-NeRF* and *RAG-Net*. Evo-NeRF, or *Evo*lving *NeRF*, builds on Instant-NGP [6], a fast implementation of NeRF, and modifies it to train concurrently to image capturing, to re-use NeRF weights between grasps and to terminate training and image collection early when sufficient task confidence if achieved. RAG-Net, or Radiance-Adjusted Grasp Network, is a network trained to compute grasps from geometry reconstructed from a NeRF.

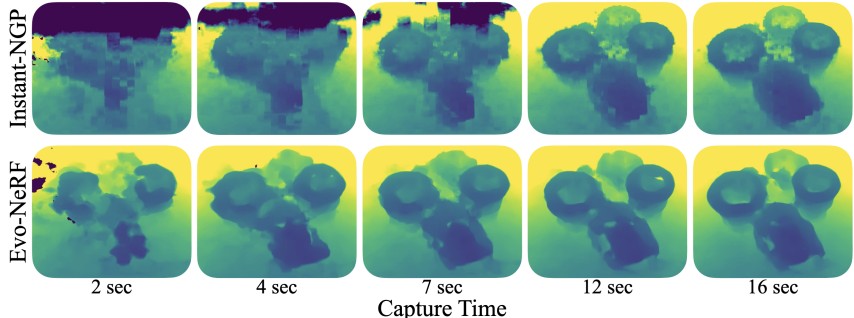

Figure 3: **Visual comparison** of Evo-NeRF's training over time vs Instant-NGP on the exact same camera trajectory. Evo-NeRF's geometry regularization improves the convergence of geometry reconstruction, resulting in fewer floaters, smoother surfaces, and ultimately faster grasps.

### 4.1 Evo-NeRF

To shorten the time to get a trained NeRF, we propose Evo-NeRF, a method that pipelines image capture with NeRF training, reuses weights in sequential grasping, adds regularization to counter effects from rapid capture, and includes an early stopping condition to start a grasp when the grasp network has high confidence.

**Image capture:** The Evo-NeRF method starts with the robot moving a camera around its workspace to capture images. Heuristically, hemispherical captures are ideal for NeRF since they maximally vary the view angles of the scene. The Evo-NeRF capture trajectory sweeps the camera through a discretized hemisphere centered at a location of interest while pointing at the center. First, the camera sweeps around the $z$-axis to maximize the variance of viewing angle early in the capture sequence. In experiments, we capture images every 3 cm while moving at 20 cm/s. A full capture trajectory takes 16 seconds and includes 80 images with trajectory shown in Fig. 1. Though images have motion blur, stopping to take each image is time-consuming and would result in fewer images, yielding lower quality reconstructions. For dataset generation in simulation, we capture 52 images per scene since we prioritize having a large variety of scenes and rendering is time consuming.

**Continual NeRF training:** NeRF training, even sped up by Instant-NGP, is a bottleneck. We propose continually training NeRF from the moment the first image is captured, and incrementally adding images to the NeRF training dataset as the camera moves to new viewpoints. This effectively pipelines the image-capture and NeRF-training processes, and allows for usable NeRF representations quickly after (and sometimes before) the capture process finishes.

During each capture motion we train NeRF in batches of 48 steps, adding new images between each batch when available. This is akin to other online neural implicit methods like iMAP [23] and NICE-SLAM [35], who also update the image sets between training batches. We compute the camera frame using the forward kinematics and pair it with each image. In practice, this yields pose error around 1 cm, which NeRF accounts for by optimizing the camera extrinsics.

**Reusing NeRF weights:** In sequential grasping scenarios, scenes often change by only the removal of the last object grasped. To take advantage the information already trained, we use the NeRF network weights from the previous grasp in the subsequent grasp. In implementation, we remove the old images from the training dataset and start capturing and training on images for the next grasp.

**Geometry regularization:** A well-known artifact of NeRF's volumetric rendering loss are *floaters*, spurious regions of density floating in space. When using NeRF for view synthesis, floaters can go unnoticed, but in grasping, floaters can lead to grasp failures. We apply 2 regularizations which increase the speed and smoothness of geometry reconstruction, visualized in Fig. 3.

First, we adapt the total-variation regularization loss (TV-loss) from Plenoxels [36] to discourage floaters and encourage smooth scene geometry. During training, at each step Evo-NeRF sample $N$ random points $p_i$ using rejection sampling to constrain samples to locations with non-trivial density values. Evo-NeRF then queries the density at all 8-connected neighbors $n_j$ at a radius $r$. The final TV-loss is $L_{\text{tv}} = \sum_{i=1}^{N} \sum_{j=1}^{8} \lambda_{\text{tv}} (\sigma(p_i) - \sigma(n_j^i))^2$, where $\sigma$ is the raw, pre-activation output from the density network, and $\lambda_{\text{tv}}$ is a loss scaling factor.

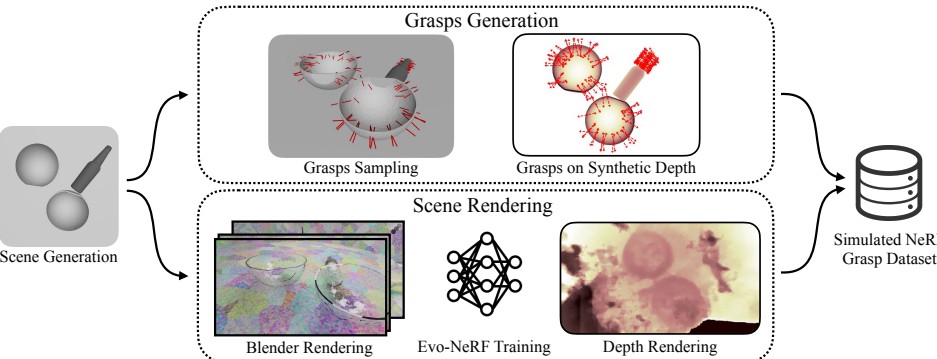

Figure 4: **Dataset Generation**. Each scene in the grasping dataset includes a subset of the training objects in simulation (Fig. 5). **Top:** Grasp generation samples grasps on the object meshes and projects them to a top-down view. **Bottom:** We render multiple views of each scene using Blender, then train Instant-NGP and render a top-down depth image. We accumulate NeRF depth rendering and projected grasps into a dataset.

Second, sampling along each ray more coarsely during training reduces floaters and quickly acquires meaningful geometry. By training with coarse samples, the NeRF is incentivized to learn a low frequency representation of the scene to minimize reconstruction error. See Appendix A.3

**Efficient perception stopping:** In scenes where NeRF is able to recover usable geometry before the full camera trajectory has terminated, Evo-NeRF can terminate the capture phase early to speed task completion. In Sec. 5.2 we present experiments showing this by querying grasp confidence of RAG-Net in a closed loop while the robot moves the camera and trains NeRF. When confidence exceeds a threshold, the capture stops early and the robot executes the grasp.

## 4.2 Grasp Planning Network

When NeRF is trained to completion with dense camera viewpoints, grasp planners trained on ground truth depth in simulation like Dex-Net [37] produce usable grasps on NeRF-rendered depth. However, in an online setting where viewpoints are limited and NeRF training terminates early, depth images rendered from NeRF appear significantly different from the ground truth depth images in simulation. To mitigate this test-time distribution shift and enable reliable grasping from online NeRFs, we train a network to predict grasps directly on NeRF-rendered depth maps.

**Network Architecture:** We train a *location* neural networks to predict the center of the grasp location when given a NeRF-rendered image; and we train a *rotation* network to predict the discretied grasp angle when given a cropped patch around the grasp location. We adapt the grasping architecture proposed by Zhu et al. [38], which suggests that an equivariant convolutional neural network learns to perform top-down grasps in fewer samples than standard networks. We train location and rotation networks on a static grasp dataset, in contrast to the online setting in Zhu et al. [38].

**Dataset Generation:** We generate the training dataset in simulation using 7 object meshes that are representative of the common household transparent objects which are graspable by the YuMi robot, shown in Fig. 5a. We model all objects with the same density as glass (2500 kg/m$^3$). We assemble scenes with labeled grasp qualities by randomly placing objects in stable poses on a planar surface and analytically sampling antipodal grasp closure axes based on mesh surface normals as in Dex-Net [39]. We use a soft point-contact model [40], and evaluate the probability of grasp success using wrench resistance [41], a common analytic measure for grasp success that is computationally inexpensive (0.02 sec per grasp) and has high precision [42]. We densely sample 1000 collision free grasps for each stable pose and use Blender to render the scenes.

**Training:** To train RAG-Net, we project sampled grasps onto the depth images and store at each pixel the maximum grasp confidence over all rotations, resulting in confidence heatmaps. We dilate and blur these heatmaps with a 3x3 kernel to smooth the predictions, and randomly augment both the depth images and the confidence heatmaps with translation, shear and scale transformations. To train the rotation network, we sample crops from grasps above 0.7 quality, and use a cross-entropy loss on the output rotation probabilities.

**Grasp Planning:** To execute a grasp from RAG-Net we render a depth image from NeRF of size $144 \times 256$ from the camera pose used during dataset generation, using the ray transmittance trun-

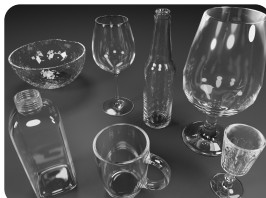 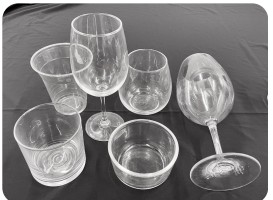 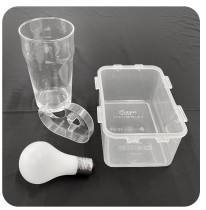 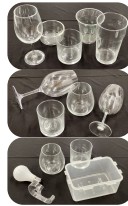

(a) Training objects (Blender) (b) In-distribution real objects (c) Out-of-dist. objects (d) Clutter

Figure 5: **Training and testing objects.** **(a)** shows Blender rendering of the 7 objects we use in data generation for computing grasps and rendering in various stable poses. Objects in **(b)** are real objects we considered in-distribution with the training objects. We also test on out-of-distribution objects shown in **(c)**. To test grasping in clutter, we setup various testing scenes with objects in and out of distribution, with examples shown in **(d)**.

|  | Dex-Net Success | RAG-Net Success | Time |
|---|---|---|---|
| Full Capture | 56 % | 89 % | 16s |
| Early Stop | 11 % | 89 % | 9.5s |

Table 1: **Single objects results:** each cell reports the average over 27 different trials. We compare success rates for full capture trajectories vs trajectories which stopped early because of sufficient grasp confidence. Early stopping results in a 41 % speed improvement with no drop in success rate for RAG-Net. Dex-Net struggles to reliably grasp on geometry rendered so early in NeRF training.

cation of Dex-NeRF [3]. We query the *location* network on this depth image to obtain a heatmap over the image of grasp confidence, then crop a patch of the depth image centered at the argmax of this heatmap. The *rotation* network takes this crop and outputs 8 grasp angle probabilities, and we take the weighted average of the argmax with its neighbors to produce the final grasp angle. We determine grasp depth by analyzing a local deprojected pointcloud from the depth image at the grasp location, and subtracting a static 1.5 cm grasp depth from the highest point.

## 5 Experiments

We evaluate the reliability of Evo-NeRF paired with RAG-Net vs Dex-Net [37], evaluate the speed improvements from early stopping captures and Evo-NeRF's reuse of weights, and ablate aspects of the system including NeRF modifications and training on NeRF depth vs ground-truth depth. We compare to Dex-Net to highlight the improvement in reliability gained from training on NeRF-rendered depth rather than ground-truth depth, and note that in Dex-NeRF [3], the NeRF model was trained for 1900x longer, with an offline, manually captured set of images with precisely calibrated poses from Colmap [43, 44]. This difference in view quality and training length from rapid capture results in a notable drop in raw Dex-Net grasp robustness because of lower quality reconstructions.

### 5.1 Physical Setup

We evaluate on a physical YuMi robot with a ZED Mini camera. The pose of the ZED relative to the arm holding it is calibrated with a chessboard once before all experiments. We surround the robot with a kitchen-like workspace containing printed images of a countertop and shelves, where test objects are positioned near the center of the workspace. The workspace has 3 LED floodlights positioned across from the robot aiming at the workspace. We use one NVIDIA GeForce RTX 3080 GPU for NeRF training and grasp network inference. We evaluate on 9 different objects, both in distribution and out of distribution with respect to the train set in Fig. 5a. We note that in general, RAG-Net performance in simulated scenes is worse than in real scenes because the synthetic dataset contains fewer camera angles than real scenes (52 vs. 80) and more difficult background textures, resulting in more floaters.

### 5.2 Rapid single object retrieval

We apply confidence-based capture early stopping (4.1) with a threshold of 70% to execute a grasp as quickly as possible as shown in Fig. 2. We place each of the 9 test objects near the center of the workspace, and report grasp success and total time spent capturing images. We repeat each

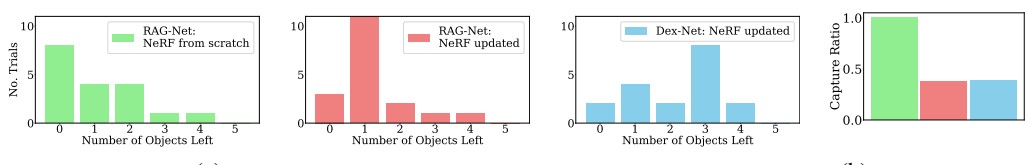

(a) Histogram of remaining objects after decluttering.

(b) Capture time ratio.

Figure 6: **Decluttering results.** **(a)** Histograms show the number of objects remaining after each trial for RAG-Net and Dex-Net (lower is better). **(b)** Reusing and updating the NeRF between grasps (red, blue) rather than recapturing the scene (green) reduces capture time by 61% (lower is better).

| | Instant-NGP | Evo-NeRF -TV | Evo-NeRF -Coarse | Evo-NeRF |
|---|---|---|---|---|
| % Trajectory Used | 80.3 % | 64.8 % | 62.0 % | **52.6 %** |

Table 2: **Ablations of Evo-NeRF regularizations.** We query Dex-Net continuously through camera capture trajectories and report the percent of the trajectory needed until the highest probability grasp is on an object. We compare vanilla Instant-NGP with Evo-NeRF, as well as ablating TV-loss and coarse ray sampling. Evo-NeRF produces successful grasps the earliest.

experiment 3 times and compare RAG-Net with Dex-Net [37] and evaluate with and without early capture stopping. Since Dex-Net does not output grasp confidence we use the same stopping point for both networks, as determined by RAG-Net. An experiment is successful if the robot grasps and places the object into the storage bin.

Table 1 summarizes the results. Using RAG-Net for early stopping results in a capture time reduction of 41 %, with no drop in reliability. On average, the robot grasps objects within 9.5 seconds with an 89% success rate over 54 trials. RAG-Net outperforms Dex-Net in grasp success by 1.6x even with a full capture of the scene as a result of its habituation to NeRF density. RAG-Net's primary failure cases are on out-of-distribution objects, specifically missing grasps on the lightbulb and tape dispenser, likely because the training set has no small profile items. In addition, some grasps failed on the sideways wineglass because it rolled out of the jaws before they closed.

## 5.3 Sequential decluttering

We evaluate on a decluttering task where multiple transparent objects are placed within 2cm in stable poses, and the robot must grasp and place all objects in the bin one by one (Fig. 1). We consider three tiers of experiment difficulties with two scenes for each tier, resulting in 6 different scenes (Fig. 5). We repeat each scene 3 times and compare against Dex-Net [37]. At the beginning of each experiment, the robot executes a full capture of the scene (Fig. 1a). After each consecutive grasp, the robot executes a much smaller capture centered at the grasp location to update the NeRF (Fig. 1c). We allow only as many grasp attempts as objects in the scene.

Results are summarized in Fig 6, showing the number of remaining objects after each trial and the speedup from updating the NeRF rather than retraining. Evo-NeRF with RAG-Net clears 72 % of test objects across all tiers while Dex-Net clears 48 % of objects. Evo-NeRF takes 39% of the capture time compared to rebuilding the NeRF from scratch with full capture trajectories after each grasp, while maintaining similar performance (76 %). This suggests Evo-NeRF retains graspable geometry over successive updates, despite their short duration. The primary grasp failure modes for this experiment are the same as in single object experiments, but sometimes the method failed to remove an object if it was moved by more than 2-3cm from accidental contact, which wasn't detected by the smaller deletion captures used between grasps.

## 5.4 Graspability ablation

We ablate the changes made to NeRF speeding geometry graspability. We capture 9 single-object and 3 multi-object scenes, then continuously train NeRF as it captures, using the same static images and holding all other hyperparameters constant. We measure the capture time needed until the highest confidence grasp output from Dex-Net lands on a real object as a proxy for graspability convergence. Table 2 shows the percent of the capture trajectory needed, and Fig. 3 shows a timelapse

of visual qualities over a capture. Results suggest that the proposed method produces graspable geometry faster, with a 32 % reduction in capture time needed to grasp using Dex-Net.

The grasp success labels used in this experiment were manually evaluated, where a human labeled a grasp as successful if its centerpoint lies on graspable geometry. To evaluate its reliability we executed grasps for single-object scenes from Evo-NeRF in the real world. All grasps labeled as successful were in fact successful (9/9), suggesting the metric used is physically reliable.

### 5.5    NeRF Depth vs Ground Truth Depth

This section investigates the distribution shift between training on ground-truth depth and testing on NeRF-rendered depth, to make the argument for training a grasp network directly on NeRF-rendered depth. We compare RAG-Net with two grasp planners: 1) Dex-Net, which is trained on a large dataset with ground-truth depth, and 2) GT-Net, which has the same architecture as RAG-Net but is trained only on ground-truth depth generated in simulation with pyrender [45]. We test on the held-out test set of NeRF-rendered depth images and report average grasp confidence using the soft-point-contact model and wrench resistance. We calibrate the grasp planners' performance by evaluating GT-Net on ground-truth depth images, a scenario with no distribution shift, and then normalize the results of all planners with respect to this performance.

GT-Net, RAG-Net and Dex-Net achieves 0 %, 42 % and 0.1% success respectively, suggesting a large distribution shift between training on ground-truth depth to testing on NeRF-rendered depth. On low quality grasps with lower than 0.1 wrench resistance, the mean depth estimation error is 2.7cm, compared to 3-5mm for values over 0.1, suggesting a failure reason here is grasping floaters.

## 6    Conclusion

We introduced Evo-NeRF, a method that rapidly captures and trains NeRFs for practical robotic grasping. While its image capture produces lower-quality reconstructions than prior work, we propose reusing trained weights in sequential grasping, geometry regularization, and continual training to obtain better 3D reconstructions. We further propose a novel training pipeline to train grasp networks on NeRF rendered depth images in simulated environments, which can predict high quality grasps in the physical environment. In experiments, Evo-NeRF and RAG-Net can grasp transparent objects within 10s of seconds with 89 % success on singulated objects.

### 6.1    Limitations and future work

RAG-Net uses rendered depth images, throwing away much of the rich 3D information present in NeRF. Future work should explore 3D grasp planner inputs from NeRF such as density voxel grids, akin to VGN [46]. While hemispherical captures are efficient for reconstructing small workspaces, it may be unsuited to tasks like finding and extracting a target object from a large scene. Though we have shown that NeRF is adaptable to geometry deletion, NeRF still resists adding new geometry because of hash collisions in the positional encoding and the density gradient being pushed towards 0 in empty regions. In our experiments we observed a failure case where an object was toppled over by the grasped object, changing the scene's geometry. Future work in adapting NeRF to changing scenes would greatly improve the practicality of real-time usage. The speed of this method is also unsuitable for industrial applications requiring sub-second cycle times, and is mainly practical for household applications such as tidying which do not have such rapid requirements.

### Acknowledgments

This research was performed at the AUTOLab at UC Berkeley in affiliation with the Berkeley AI Research (BAIR) Lab and the CITRIS "People and Robots" (CPAR) Initiative. The authors were supported in part by donations from Google, Siemens, Toyota Research Institute, and Autodesk and by equipment grants from PhotoNeo, NVidia, and Intuitive Surgical. This material is based upon work supported by the National Science Foundation Graduate Research Fellowship Program under Grant No. DGE 2146752. Any opinions, findings, and conclusions or recommendations expressed in this material are those of the author(s) and do not necessarily reflect the views of the sponsors or National Science Foundation.

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

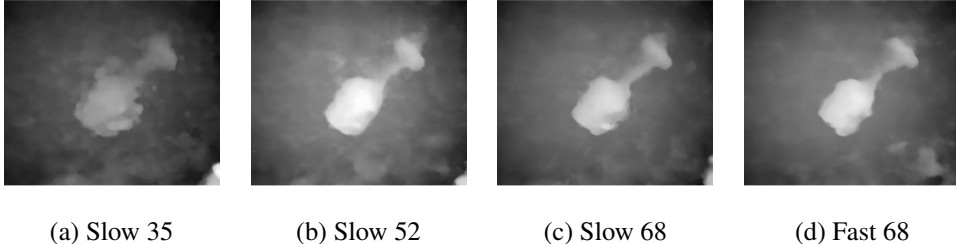

| (a) Slow 35 | (b) Slow 52 | (c) Slow 68 | (d) Fast 68 |

Figure 7: **Motion Blur.** Top-down depth images rendered using models trained on different datasets. We capture the *Slow* datasets (a, b and c) from fixed views while the camera is held statically, and the *Fast* dataset (d) while the camera is in motion.

## A  Evo-NeRF

### A.1  NeRF and scene parameters

We use the default Instant-NGP parameters except for the following changes: (1) we use a learning rate of 0.02 instead of 0.01 (2) we use a hash table size of $2^{17}$ feature vectors instead of $2^{19}$ (3) we use a density network with 16 hidden neurons instead of 64. These changes made small improvements in geometry learning speed. We also increase the frequency of extrinsic optimization gradient steps (n_steps_between_cam_updates parameter) to every step, improving the speed of convergence on noisy camera poses added during arm motion. The scene bounds ("aabb_scale") are set to a 2 meter cube to fit the entire workspace inside, with a scene scale of 1.0. We set the near distance for raymarching during NeRF training to be 0, to avoid missing objects if they are close to the camera.

### A.2  Capture trajectory

The full capture trajectory is centered at the center of the workspace, with $\theta$ values ranging from $85°$ to $75°$ ($\theta$ rotates about the z axis upwards from the table, such that x points away from the robot). The $\phi$ range, which describes inclination from the table surface, goes from $15°$ to $50°$. The arm makes 3 sweeps about the z axis, linearly varying the $\phi$ value between the range on each sweep, as visualized by the red arrow in Fig. 1. We use 1280x720 images, with a whitebalance and exposure which are held static after an auto-calibration from the camera.

### A.3  Evo-NeRF parameters

For TV-regularization, we sample $N = 256000$ points at each iteration with a rejection sampling threshold of 0.01 for minimum local density. The sampling radius $r$ we use is 0.3mm, and the loss scaling $\lambda_{\text{tv}}$ is $\frac{15}{N}$. TV-loss is implemented as a set of CUDA kernels for speed, resulting in only about a 10% slowdown of training.

In Instant-NGP the distance between samples grows proportional to distance along the ray, and we scale this distance to be 10x larger. To implement coarser ray sampling, the value of the parameter which controls sample acceleration, ("cone_angle_constant" in Instant-NGP) is 0.04, up from the default value of 0.004.

### A.4  Motion Blur

In this experiment we investigate the effect of motion blur on the quality of the top-down depth image rendered with Evo-NeRF. We qualitatively compare depth images rendered using models that were trained with four different datasets: *Slow 35*, *Slow 52* and *Slow 68* that were trained on 35, 52 and 68 images captured while the camera was static, and *Fast 68* trained on 68 images captured while the camera was in motion (as described in Sec 4.1), all within 1.5 cm between the *Fast* and *Slow* datasets. Fig 7 shows the top-down rendered depth images. Results suggest that the depth quality is higher when the model is trained on a large dataset, and that there is no significant difference in depth quality between the *Fast* and *Slow* capturing methods. Though motion blur has

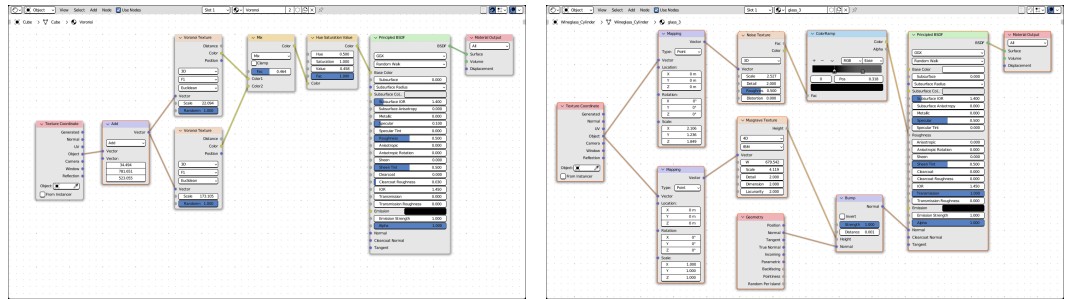

Figure 8: Random worksurace texture (left) and glass texture (right) in blender. We use textures to randomize the background and simulate imperfection in glass.

a known negative affect on NeRF quality[47], for the speed our arm moves at this appears to not be a significant problem.

# B  Dataset generation

We choose the 7 object meshes based on the three criteria: (1) likely to be made of glass, (b) fit within the workspace of YuMi, (c) has a watertight mesh with outward-facing surface normals.

For grasp generation, we calculate the stable pose orientations of each mesh and rank them by their quasi-static probabilities using Trimesh [48]. Based on the ranking, we select the top 10 stable poses to sample 1000 grasps. We ignore stable poses where no grasp exists. We analytically calculate grasp success via robust wrench resistance [49]. We perturb the grasp pose with small translation noise (from a normal distribution with $\mu = 0$, $\sigma = 0.003$ m) and small rotation noise (from a normal distribution with $\mu = 0$, $\sigma = 0.003$ rad) and calculate wrench resistance on 10 samples for estimating the grasp success probability. We create multi-object scenes based on single-object scenes. To do so, we sequentially sample objects on different stable poses and randomly generate a SE(2) transform that will not collide with the objects that have already been placed on the planar surface. We define the $z$–axis as normal and pointing outward from the worksurface. We sample $x$ and $y$ position from a uniform distribution between $\pm 0.2$ m and the z–axis rotation from a uniform distribution between $\pm \frac{\pi}{2}$. We reuse the grasps sampled for single object scenes and filter out the grasps that are in collision. We check collision between objects and grasps with the Flexible Collision Library [50]. We meshify the grasps for collision checking by using a YuMi gripper mesh model under the rigid transform given by the grasp pose.

To reflect the reachable workspace of the robot for capturing images, we record the camera intrinsics and 52 camera poses along the image capture trajectory with the physical robot. We use fewer views during dataset generation than physical capture trajectories to speed Blender rendering. For each of the simulated environments, we render images at the recorded camera poses with small translation noise ($\pm 5$ mm) and rotation noise ($\pm 5°$). We randomize the number of lights between 1 and 5, light location, and total wattage. To speed up rendering, we reduce the numbers of rays cast to a minimum level to achieve realistic renders, and use CUDA-based renderer in Blender.

To randomize background and simulate real-world imperfections found in glass, we use two textures in Blender (Fig. 8). We use a randomized texture for the background consists of two blended random "Voronoi" nodes to produce both high and low frequency patterns. For the glass texture, we create a transparent material with an index of refraction that matches glass and many plastics. We also add random textures to simulate hazy glass and scratches. Prior work observed that NeRF performed better on real-world glass than simulated glass, observing that simulated glass had no imperfections.

We then train a NeRF model for each scene for 1000 steps, comparable to the number used on the robot in real-time, and render depth images from NeRF. We also generate ground-truth depth images using Pyrender [45] for each scene, using the same camera extrinsics as the NeRF rendered depth image. In experiments, each scene has between 1 and 3 objects and there are a total of 8667 distinct scenes, 237 held out as a test set.

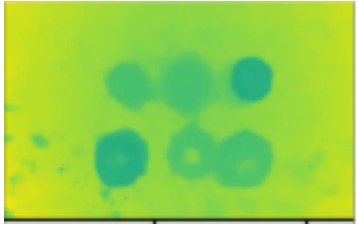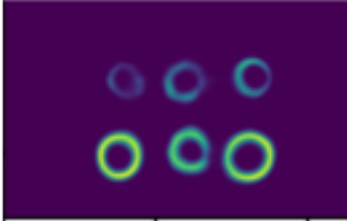

Figure 9: Heatmap output for objects including upside-down glasses. In this scene the top 3 objects are upside-down (and hence ungraspable) and the bottom 3 are graspable. The left image shows depth rendered from NeRF and the right image shows the location heatmap output by RAG-Net. Note how the heatmap activates much less on upside-down objects (<15% confidence) compared to graspable glasses (80% confidence).

## C  RAG-Net

### C.1  Depth rendering

When rendering depth from NeRF, we use a minimum transmittance threshold of 0.9, which means that rays which have passed through a total of 0.1 density terminate. This extra sensitivity is to allow perceiving depth from transparent objects. Because density has physical meaning, in practice the value of this parameter is reusable across all scenes, in our experience not requiring tuning. During depth rendering we ignore density which exists more than 35cm above the workspace surface, a value 2x larger than the largest test object, to help in removing floaters far above the scene.

### C.2  Architecture details

The architecture we use is identical to Zhu et al. [38], except for adding an additional fully-connected layer at the output of the rotation prediction network to be more agnostic to input patch sizes. The location prediction network uses an equivariant U-Net architecture, and the rotation network is a 9-layer equivariant ResNet. The rotational equivariance operates on the cyclic group $C_8$ for the location prediction network and on the quotient group $C_{16}/C_2$ for the rotation network as top-down grasp rotation is invariant to rotations by $\pi$ radians.

### C.3  Training details

We use PyTorch Lightning for training, with a batch size of 64 for the rotation network and 32 for the location network. We use the Adam optimizer with learning rate 1e-3 and weight decay 1e-5. Models are trained for 100 epochs with an exponential learning rate decay of 0.994. The patches used as input to the rotation network are augmented by 5 pixels of random translation, and the location depth images are augmented by 10% translation, $\pm 5°$ shear, and a scale range of $80\%$ to $100\%$.

### C.4  Upside down glasses

In all of our experiments we test on graspable, upright objects. So, a natural question is whether RAG-Net has learned a trivial grasp function, like grasping at the edge of any round object. To answer this question we explore what RAG-Net outputs on ungraspable, upside-down objects to sanity check its output. To do this we run a decluttering task with 3 upright and 3 upside down glasses, and inspect the confidence outputs on upside down glasses compared to upright. Given 3 actions, the system correctly removes the 3 graspable glasses and leaves the upside-down ones untouched. Fig 9 shows RAG-Net's output on this scene before the first grasp. Although the upside down glasses have ring-like heatmap outputs similar to upright cups, the highest activation on upside down glasses is 15%, which suggests that RAG-Net seems to have learned a non-trivial grasp function. Ideally, confidence on impossible grasps would be near 0, a shortcoming that could perhaps be a result of an imbalanced dataset, where more objects are upright than upside down. Cultivating a dataset with equal numbers of graspable and ungraspable poses could address this issue.

| Objects | DexNet | | RAG-Net | |
|---|---|---|---|---|
| | Early Stop | Full Capture | Early Stop | Full Capture |
| Wineglass Upright | 0/3 | 3/3 | 3/3 | 3/3 |
| Whiskey Glass | 0/3 | 1/3 | 3/3 | 3/3 |
| Wineglass Sideway | 0/3 | 2/3 | 2/3 | 2/3 |
| Plastic Cup | 0/3 | 0/3 | 3/3 | 3/3 |
| Bowl | 0/3 | 2/3 | 3/3 | 3/3 |
| Tape Dispenser | 3/3 | 3/3 | 1/3 | 2/3 |
| Square Bowl | 0/3 | 1/3 | 3/3 | 3/3 |
| Tall Glass | 0/3 | 3/3 | 3/3 | 3/3 |
| Light Bulb | 0/3 | 0/3 | 3/3 | 2/3 |
| Average | 11% | 56% | 89% | 89% |

Table 3: Detailed single object retrieval results. For each object, the experiment is repeated 3 times. The number show the success grasps out of the 3 grasps. The last row show the average success grasp over all objects. Note that all of RAG-Net's failures come from the sideways wineglass, tape dispenser, and lightbulb. The latter two suffer in performance because they are highly out of distribution shaped objects, and the former experiences a 66% success rate because grasp precision is much more important for grasping the stem or base, where a small pose error can knock the wineglass out of position.

| Scene (N objects) | Dex-Net, NeRF updated | RAG-Net, NeRF from scratch | RAG-Net, NeRF updated |
|---|---|---|---|
| 0(4) | 1,4,3 | 4,4,4 | 4,4,4 |
| 1(5) | 1,4,2 | 3,4,4 | 4,4,4 |
| 2(4) | 1,3,4 | 3,4,0 | 0,3,3 |
| 3(4) | 1,3,0 | 2,4,2 | 3,3,3 |
| 4(5) | 2,3,3 | 4,3,2 | 4,2,4 |
| 5(4) | 1,1,1 | 4,4,4 | 3,2,3 |

Table 4: Detailed decluttering results. Each scene is repeated 3 times and the method is given as many grasp attempts as the number of objects. Numbers in the parenthesis show the number of objects in this scene. Numbers in the table show the number of objects extracted after all actions finish (higher is better). Scene 2 seems to be an outlier in performance for RAG-Net, with 2 runs where no objects were cleared. This is due to a specific wineglass which RAG-Net consistently collided with during grasps, resulting in an early failure for the trial.

## D    Experiments details

### D.1    Single object

Table 3 reports the per-object success for RAG-Net and Dex-Net on early-stopped and full capture trajectories, along with a discussion of their implications in the caption.

### D.2    Decluttering

Table 4 reports per-scene success for all scenes for Dex-Net and RAG-Net on Evo-NeRF as well as training NeRF from scratch, along with a discussion of the results in the caption.

