# OpenReview forum: "Evo-NeRF: Evolving NeRF for Sequential Robot Grasping of Transparent Objects"
_robot-learning.org/CoRL/2022/Conference — CoRL 2022 Oral_

### Official Review · Reviewer_UJkS · 2022-07-08

**Originality:** Good
**Technical Quality:** Very Good
**Clarity Of Presentation:** Very Good
**Impact:** 2

**Recommendation:**

Weak Accept: I recommend accepting the paper, but will not argue for my recommendation if the majority of other reviewers have a different opinion.

**Summary:**

The authors propose a method for robotic grasping of transparent objects combining a NeRF-based perception and rendering module with a CNN-based grasp network. To reduce the long training time required by previous publications on grasping transparent objects with NeRF-based perception, the authors propose several improvements:
- They leverage a recent NeRF-variant, which reduces training time drastically
- They train on images captured during robot motion and use a predicted confidence score to decide when to stop the capturing phase
- They re-use weights of previously trained NeRFs for faster training in evolving scenes
Furthermore, to cope with the inaccuracies resulting from capturing during motion, the authors propose a geometric regularization.

**Issues:**

Section 5.4:
- Did the authors test, whether the graspability heuristic used here is actually reliable, i.e. whether the predicted grasps actually lead to successfully grasped objects?

Section 5.5:
- Why are different numbers of views used in simulation and in the real setup? Adding additional views in simulation should be cheap.

Experiments:
It would be interesting to see, how a method capturing less images from fixed view points performs as compared to the proposed method capturing images while moving.

Minor:
Do the authors have any intuition of why their method cannot cope with adding objects to the scene?

**Quality Of The Limitations Section:**

Additional details required

**Reviewer Expertise:**

4: The reviewer is confident but not absolutely certain that the evaluation is correct

**Robotics Focus:**

Sufficient demonstration on hardware

**Strengths And Weaknesses:**

Strength:
- The method achieves a significant reduction in training time compared to previous methods

Weaknesses:
- Even though the grasp inference time has been reduced drastically compare to previous methods, the proposed approach is still too slow for practical applications. This should be discussed in the limitations.
- There are some unclarities listed below in the issues.

**Summary Of Recommendation:**

The suggested method drastically improves inference time for robotic grasping of transparent objects. The improvements the authors propose compared to previous work seem interesting and effective. Though the overall system is probably still too slow for real world use cases, the proposed method is a good step towards that goal.

---

> ### Author Response · Authors · 2022-08-20
> **Response to reviewer UJkS**
>
> Thank you for noting that “The method achieves a significant reduction in training time compared to previous methods”.  Please see responses to concerns below.
>
> - “the proposed approach is still too slow for practical applications”
>
> We have revised the paper in the introduction to clarify potential applications such as home and kitchen grasping eg unloading a dishwasher, for which 10 seconds per grasp is often practical.  This timing may also be practical for some applications in retail, restaurant, and healthcare, but we fully agree that the method is not sufficiently fast for industrial applications such as assembly and have added this point to the Limitations section.
>
> - “Did the authors test, whether the graspability heuristic used here is actually reliable, i.e. whether the predicted grasps actually lead to successfully grasped objects?”
>
> Thank you for asking about this. Last week we ran the following experiment: we captured the same scenes used in the ablation with Evo-NeRF, then executed the heuristically chosen grasp in the real world to verify its success. In this experiment, every grasp selected by the heuristic as a success was successful in the real world (9/9), and 2 scenes never triggering the heuristic graspability condition. We have included this in the revised section 5.4.
>
> - “Why are different numbers of views used in simulation and in the real setup? Adding additional views in simulation should be cheap”
>
> We revised the paper in sections 4.1 and 5.1 to clarify that with our computing resources, we found that rendering photo-realistic views in simulation is time consuming when done at large scale. When we generated the training dataset for Rad-Net we emphasized a large variety of scenes with fewer views per scene (52 views) than having fewer scenes with more views. In physical experiments we didn’t have the same limitation and so we collected additional views (80 views).
>
> - “It would be interesting to see, how a method capturing less images from fixed view points performs as compared to the proposed method capturing images while moving”
>
> We have revised the paper to note this in Section 4.1.  In our experience, the number of views required makes statically-mounted cameras or pausing to capture each image impractical (due to cost and speed respectively), despite providing better image quality without motion blur. In addition, NeRF is more sensitive to the number of views (denser is better) than the image quality (motion blur), so using more viewpoints with motion blur results in better reconstructions than a few viewpoints without blur. We will run an experiment capturing the same scene with sparsely laid out viewpoints from the camera at rest to investigate the difference in reconstruction quality.
>
> - “Do the authors have any intuition of why their method cannot cope with adding objects to the scene?”
>
> This is an interesting question!  We revised the paper in the Limitations section with a discussion about this.   Removing objects is much easier than adding objects due to two key details in NeRF and Instant-NGP:
>
> 1) The volumetric loss function used to train NeRF causes regions with near 0 density to disappear from the gradient, so adding density back to previously empty regions takes many more optimization steps.
>
> 2) Instant-NGP uses a static hash function to map 3D coordinates to feature vectors. A side effect is that in a trained NeRF, the feature vectors corresponding to empty space are dominated by non-empty regions of the scene that hash to the same index. To learn new geometry, these shared feature vectors need to change to represent the new geometry, which takes longer when adding geometry vs removing.
>
> Intuitively, removing  geometry from a scene is easier than adding because it requires learning no additional color information; only removing density from the scene, whereas adding geometry requires learning both additional density and viewing-angle dependent color.

---

> ### Author Response · Authors · 2022-08-24
> **Response to reviewer UJkS - additional results**
>
> ## “It would be interesting to see, how a method capturing less images from fixed view points performs as compared to the proposed method capturing images while moving”
>
> Following this suggestion, we ran 4 experiments: in the first 3 the images were captured from fixed view points (35, 52 and 68 view points) and in the last the images were captured while moving (68 view points).
> The results suggest that the depth quality is higher when the model is trained on a large dataset, and that there is no significant difference in depth quality rendered from models trained on datasets captured from fixed viewpoints or while moving.
> We added results to the appendix (A.4).

---

> ### Comment · Reviewer_UJkS · 2022-08-25
> **Response to authors**
>
> I thank the authors for their thorough replies and clarifications. I also appreciate the additional details and experiments the authors provide.

---

### Official Review · Reviewer_NJjN · 2022-07-31

**Originality:** Good
**Technical Quality:** Good
**Clarity Of Presentation:** Excellent
**Impact:** 3

**Recommendation:**

Weak Accept: I recommend accepting the paper, but will not argue for my recommendation if the majority of other reviewers have a different opinion.

**Summary:**

This manuscript introduces a method to progressively update and quickly train NeRF models to grasp transparent objects. The authors propose to reuse weights for sequential grasping tasks, use an early stopping strategy to speed up the completion, and add regularization loss to mitigate the floaters issue. To reduce the sim2real gap, the authors propose to train the network to predict grasps directly on NeRF-rendered depths maps. The proposed pipeline is applied in real-world experiments, achieving an 89% success rate for single object grasping and reducing the capture time significantly for both single object grasping and sequential grasping tasks.

**Issues:**

1. In lines 136-138, “In sequential grasping scenarios, scenes often change by only the removal of the last object grasped. To take advantage the information already trained, we use the NeRF  network weights from the previous grasp in the subsequent grasp.” In the sequential grasping scenarios, some objects might move (such as translate or even rotate) due to collision, is the strategy of reusing NeRF weights still useful?

2. In lines 144-145, “we adapt the total-variation regularization loss (TV-loss) from Plenoxels [34] to discourage floaters and encourage smooth scene geometry” Does the regularization loss result in larger depth estimation error, thus decreasing the grasp performance if the object surfaces are not smooth?

3. In line 237, “ If a grasp planner generates a wrong  grasp leading to a joint over-torque error, we terminate the experiment as a failure.” It will strengthen the paper if the failure cases are also contained in the real-world video.

4. In line 259, “GT-Net achieves 0 %, Rad-Net achieves 42 %, and Dex-Net achieves 0.1%, suggesting that there is a  large distribution shift from training on ground-truth depth to testing on NeRF-depth in simulation.” What is the error between predicted depth maps and ground-truth depth maps in these grasping regions? It will strengthen the paper if failure case analyses are provided.”

**Quality Of The Limitations Section:**

Limitations are addressed clearly

**Reviewer Expertise:**

4: The reviewer is confident but not absolutely certain that the evaluation is correct

**Robotics Focus:**

Sufficient demonstration on hardware

**Strengths And Weaknesses:**

Strengths:
1. The paper is well written and easy to follow.
2. It’s exciting to see that the proposed grasping pipeline only takes seconds to grasp a transparent object with the NeRF model.

Weaknesses:
1  It’s not clear whether the proposed strategy of regularization is generally applicable or not (See Issue 2).
2. More failure case analyses and real-world videos of these failure cases actually strengthen the paper.


**Summary Of Recommendation:**

The manuscript introduces a pipeline for robots to grasp transparent objects taking only seconds per object, which is useful and important for many real-world robotic applications. But it’s not clear whether the proposed strategies are generally helpful or not (Issue 2) or only work with specific cases (Issue 1). Therefore, I will recommend weak acceptance.

---

> ### Author Response · Authors · 2022-08-20
> **Reply to reviewer NJjN**
>
> Thank you for your comment that “It’s exciting to see that the proposed grasping pipeline only takes seconds to grasp a transparent object with the NeRF model.” Below we respond to each of your concerns:
>
> - “In the sequential grasping scenarios, some objects might move (such as translate or even rotate) due to collision, is the strategy of reusing NeRF weights still useful?”
>
> Thank you for noting this and we’ve added a discussion about this to the Limitations Section and a case study where this leads to failure in Section 5.3 in the revised version.  It is possible for objects to move during extraction. In our experiments we minimized this by placing  objects in stable poses 2 cm apart from neighboring objects. In future work we will explore how we might detect such changes and reset the NeRF computation accordingly.
>
> - “Does the regularization loss result in larger depth estimation error?”
>
> Thank you for your  insightful point that regularizing the 3D density may smooth fine details that could potentially affect the grasp success on objects with complex geometry. We acknowledge this in section 4.1, noting many household/lab transparent objects have smooth geometry, and thus are not affected by this potential issue.
>
> - “It will strengthen the paper if the failure cases are also contained in the real-world video.”
>
> We fully agree that a more in-depth treatment of failure cases will improve the paper and we have revised the Results section accordingly.  We will also include failure cases in all videos.
>
> - “What is the error between predicted depth maps and ground-truth depth maps in these grasping regions?”
>
> We appreciate your suggestion on evaluating depth error.  We ran a new experiment as you suggested: first we evaluate depth-map accuracy on all grasp locations output from Rad-Net, then binned results into grasp quality buckets. In a test set composed of 465 scenes, we find that for low quality grasps under .1 (as evaluated by the wrench resistance metric), the depth error is on the order of 2.7cm, while for high quality grasps where the wrench resistance Metric is .5-1.0,  the depth error is 3-5mm. This suggests that a common failure case for this experiment is grasping near floaters where no actual geometry exists. We added this in Section 5.5 of the revised paper and will add all details in the Appendix.

---

### Official Review · Reviewer_iTZS · 2022-08-01

**Originality:** Very Good
**Technical Quality:** Very Good
**Clarity Of Presentation:** Good
**Impact:** 4

**Recommendation:**

Strong Accept: I recommend accepting the paper and will argue for my recommendation even if other reviewers hold a different opinion.

**Summary:**

This paper studies the problem of grasp synthesis of transparent objects using RGBD images. The authors propose a pipeline based on Neural Radiance Fields (NeRFs), which they modify in several ways to make the training process amenable to real-time robotics use. These include interleaving dataset updates with NeRF training, additional loss terms and hyperparameter adjustments, and early stopping. They further propose to train a grasp planing network (based on prior work) directly on the NeRF depth images, rather than the images generated by the depth sensor. The authors study their method's performance onboard a physical robot, and demonstrate much faster performance than DexNet, fast re-training of the NeRF for decluttering, and ablate the proposed NeRF modifications to clearly demonstrate their contribution.

**Issues:**

- I'm not sure the Sim2Real gap is discussed in enough detail in the text. Section 5.5 seems to address this, but mostly reports that training grasp networks on RGB-D images from reflective objects (this is the motivation of the paper?). I would argue a better experiment would be to test this on opaque, matte objects -- if the simulation-trained grasp planner performs similarly to DexNet in this case, you have made a strong case that the Sim2Real gap is quite small.

- In general, the text in the experiments section suffers from a lack of clarity (Section 5.5 was particularly hard to parse). I think this part of the paper would benefit from some editing for clarity/grammar.

- This is more a direction for future work, but it seems there would be better ways to generate the capture trajectory based on error metrics rather than following a path on a sphere.

- Similarly, I found the "deletion" mechanism a bit less-than-compelling. This isn't an issue that should prevent publication, but it seems a more elegant way to handle the problem would be to segment the NeRF into several smaller objects that are composited during rendering.

**Quality Of The Limitations Section:**

Additional details required

**Reviewer Expertise:**

5: The reviewer is absolutely certain that the evaluation is correct and very familiar with the relevant literature

**Robotics Focus:**

Sufficient demonstration on hardware

**Strengths And Weaknesses:**

+ The proposed method can quickly train a NeRF representation, and works for transparent objects, a current open problem in grasp synthesis from visual data.

+ The hardware experiments are thorough and demonstrate a clear improvement over (approximately) the state of the art, DexNeRF. The modifications to the NeRF training, in particular, are novel and appear quite useful.

- The method requires a somewhat prescribed "capture trajectory" to build the NeRF model, although this is usually stopped early once the grasp generation is confident.

- The grasp planning network has a Sim2Real issue (trained in sim, deployed on real objects) although the authors claim training on NeRF depth ameliorates this somewhat.

**Summary Of Recommendation:**

Strong accept. The proposed modifications to the NeRF training make it applicable for real-time use in robotics, and are ablated to demonstrate their benefits in real hardware. Grasp synthesis for transparent/reflective objects is still quite challenging, and the authors present a method that seems to do this quite effectively. These are exciting, compelling, and novel results.

---

> ### Author Response · Authors · 2022-08-20
> **Reply to reviewer iTZS**
>
> We appreciate the strongly positive review and conclusion that “These are exciting, compelling, and novel results.” Regarding the additional points:
> - “The grasp planning network has a Sim2Real issue (trained in sim, deployed on real objects) although the authors claim training on NeRF depth ameliorates this somewhat. I would argue a better experiment would be to test this on opaque, matte object”
>
> Thank you for raising this point. There are two distribution shifts: 1) between simulated NeRF and real-world NeRF, and 2) between training on ground-truth depth and testing on NeRF-rendered depth. We have revised the wording in the Intro and Sections 5.1 and 5.5 to clarify this distinction and that our focus is on the latter,  which can be mitigated by training on NeRF-rendered depth. Interestingly, performance of the grasping network is significantly worse in simulation because the simulated scenes have a more difficult background texture and fewer views, resulting in more floaters.
> Thank you for your suggestion to evaluate with opaque objects.  In physical trials, we found that it is very hard to obtain accurate ground truth depth information, since a small difference between a depth camera and the NeRF rendered view, as well as the position of objects can affect the grasp predictions, but we will continue to think about this.
>
> - “In general, the text in the experiments section suffers from a lack of clarity”
>
> We revised section 5.5 to be much more clear by describing the distribution shift between ground truth depth and NeRF-rendered depth, and clarifying details on the experiments we ran to evaluate it. In addition we also revised the captions of figures and tables in the experiments section, and reworded the results to be clearer.
>
> - “it seems there would be better ways to generate the capture trajectory based on error metrics”
>
> This is a good point that we clarified in the revised version. We have initial information, such as a prior on the location of objects, so we tune the image capture trajectory to emphasize this location. We found that for a sequential grasping task, where the goal is to remove objects one by one from the workspace, a path along the sphere was efficient as it rapidly varies the viewpoint of the camera while being kinematically smooth. We agree with the reviewer’s point that this approach implicitly assumes priors on the object location and size and may be unsuited to tasks like extracting a target object from a heavily occluded scene, or searching for an object in a large workspace. We added a discussion of this topic to the future work and limitation sections
>
> - “Similarly, I found the "deletion" mechanism a bit less-than-compelling[…] it seems a more elegant way to handle the problem would be to segment the NeRF into several smaller objects that are composited during rendering.”
>
> Thank you for this interesting suggestion! Compositing NeRFs from multiple objects could be a very good way to rapidly update the scene geometry, for example taking inspiration from methods like Control-NeRF [1]. This direction poses some challenges: 1) determining dynamically which regions of the scene should be broken into different NeRFs may be error-prone when objects are very close to each other, 2) but strong priors on object geometry may facilitate segmentation, and 3) training a composite NeRF from scratch may sacrifice speed of training relative to reusing a single NeRF.
>
> In this paper, we evolve one  NeRF over time primarily because of its simplicity and effectiveness at object removal but we added a note about segmentation to the Future Work section.
> [1] https://arxiv.org/abs/2204.10850

---

### Meta-Review · Area_Chair_Jfiq · 2022-08-09

**Recommendation:** Accept (Oral)
**Confidence:** 5

**Metareview:**

The paper introduces a new method for sequential robotic grasping by learning NeRF models of a scene during grasping. The novelty of the method is on how to train NeRF models faster during grasping as well as a grasping network adapted to images generated from NeRF models. Overall, the proposed method is novel and solves an important grasping problem in robotics. The experimental results demonstrate the effectiveness of the proposed method. Concerns from the reviewers are about unclear parts in the paper.

The concerns from the reviewers have been successfully addressed during the rebuttal. The authors are encouraged to revise the final paper accordingly.

**Best Paper Nomination:**

No

---

> ### Author Response · Authors · 2022-08-28
> **Reply to Meta Reviewer**
>
> Thank you for serving as area chair for our paper. We have made the following major updates to our submission:
>
> * As requested by reviewer UjkS, we have tested and verified the reliability of the graspability heuristic through physical experiments, and compared the performance of the model when trained on images from fixed views.
> * As requested by reviewers iTZS, NJjN and UjkS, we have revised Sections 5-6 of the paper with additional details about the experiments we ran, the failure cases and future work.
> * As requested by reviewer iTZS, we added a discussion on the capture trajectory and clarified the distribution shift this method addresses.
> * As requested by reviewer NJjN, we added new results and analysis regarding the depth error near grasping regions.
> * As requested by reviewer UjkS, we clarified potential applications of the method in the introduction section.
> * As requested by reviewer UjkS, we added a discussion to the limitations section regarding the difference between removing and adding objects to the scene.